# Acute kidney injury and its predictors among HIV-positive patients in Africa: Systematic review and meta-analysis

**Abere Woretaw Azagew** [ORCID] *, **Hailemichael Kindie Abate, Yohannes Mulu Ferede, Chilot Kassa Mekonnen**

Department of Medical Nursing, School of Nursing, College of Medicine and Health Sciences, University of Gondar, Gondar, Ethiopia

* wabere@ymail.com

## Abstract

### Background

cute kidney injury(AKI) is a rapid loss of the kidney's excretory function, resulting in an accumulation of end products of nitrogen metabolism. The causes of AKI in HIV-positive patients are not well investigated, but it may be associated with antiretroviral drug side effects and HIV itself. Even though there were studies that reported the prevalence of AKI among HIV-positive patients in Africa, their findings were inconsistent across the studies.

### Methods

We searched on PubMed, Embas, Ebsco, OVID, Cochrane Library, and other supplementary search engines, including Google and Google Scholar. Articles published upto July 2023 were included in this review study. The quality of the study was assessed using the Newcastle-Ottawa Scale for cross-sectional, case-control, and cohort studies. The data were extracted using a Microsoft Excel spreadsheet and exported to Stata version 14 for analysis. A random effect meta-analysis model was used to estimate the pooled prevalence of AKI among HIV-positive patients. Heterogeneity was evaluated using Cochrane Q statistics and I squared ($I^2$). Furthermore, the graphic asymmetric test of the funnel plot and/or Egger's tests were computed to detect publication bias. Sensitivity analysis was computed to see the effect of a single study on the summary effects. To treat the publication bias, a trim and fill analysis was carried out. The protocol of this review has been registered in an international database, the Prospective Register of Systematic Reviews (PROSPERO),with reference number CRD42023446078.

### Results

A total of twenty-four original articles comprising 7913HIV-positive patients were included in the study. The pooled prevalence of AKI among HI-positive patients was found to be 23.35% (95% CI: 18.14–28.56%, $I^2$ = 97.7%, p-value <0.001). Low hemoglobin (Hgb <8mg/dl) was found to be the determinant factor for AKI among HIV-positive patients (AOR = 2.4;

**Data Availability Statement:** All relevant data are within the paper and its Supporting information files.

**Funding:** The author(s) received no specific funding for this work.

**Competing interests:** The authors have declared that no competing interests exist.

**Abbreviations:** AKI, Acute Kidney Injury; AOR, Adjusted Odds Ratio; CI, Confidence Interval; HIV, Human immunodeficiency virus; PRISMA, Preferred Reporting Item for Systematic Review and Meta-Analysis; WHO, World Health Organization.

95% CI:1.69–3.4, $I^2$ = 0.0%, p-value = 0.40). In meta-regression analysis, sample size was the possible source of variation among the included studies (AOR = 3.11, 95%CI: 2.399–3.83).

## Conclusions

The pooled prevalence of AKI among HIV-positive patients was high. HIV-positive patients with low hemoglobin levels are at risk of developing AKI. Hence, regular monitoring of kidney function tests is needed to prevent or delay the risk of AKI among HIV-positive patients. Healthcare workers should provide an integrated healthcare service to HIV-positive patients on the prevention, treatment, and reduction of the progression of AKI to advanced stages and complications.

## Background

The human immunodeficiency virus (HIV) is still the leading challenge worldwide [1, 2]. Globally, 39 million people were living with HIV at the end of 2022. Africa remains the most severely affected region, with nearly one in every twenty-five adults living with HIV and accounting for more than two-thirds of the people living with HIV worldwide [3].

Renal dysfunction, especially acute kidney injury (AKI), is an important cause of hospitalization and mortality among HIV-positive patients. It is a common complication in HIV-positive patients [4]. Injury or diseases, including HIV infection, can damage the kidneys and lead to kidney diseases. In people with HIV, poorly controlled HIV infection and co-infection increase the risk of AKI [5].

The exact causes of HIV-associated AKI are not well investigated, but it is associated with volume depletion, sepsis, and the intake of nephrotoxic medications [6]. It is also associated with individual risk factors, HIV-correlated factors, and antiretroviral drug toxicity [7]. AKI can lead to life-threatening complications such as end-stage renal failure, volume overload, electrolyte disturbance, and multi-organ dysfunction [8].

AKI can affect the entire population, but its severity is higher among HIV-positive patients. The death rate of AKI among HIV-positive patients was 21.2%, 25%, and 35.3%, compared with HIV-negative individualsat10%, 23.3%, and 16% at one, two, and five-year follow-up periods [9]. HIV-positive patients with advanced disease, advanced age, pre-existing kidney diseases, and concomitant use of nephrotoxic medication are at increased risk of adverse renal events [10].

In Africa, even though there were different studies associated with acute kidney injury among HIV-positive patients, the findings were inconsistent across the studies. Therefore, this study aimed to estimate the prevalence and identify predictors of AKI among HIV-positive patients in Africa.

## Methods

### Study protocol registration and reporting

The protocol has been registered on PROSPERO with reference number CRD42023446078. The reporting of this review follows the Preferred Reporting Item for Systematic Review and Meta-Analysis (PRISMA-2020) checklist [11] (S1 File).

## Study design and search strategies

We searched on PubMed, Embas, Ebsco, OVID, Cochrane Library, and other supplementary search engines: Google and Google Scholar. Endnote Version 7 reference management software was used to download, organize, review, de-duplicate, and cite the articles. Our comprehensive search strategies were carried out using controlled vocabularies such as medical subject headings(MeSH) terms. Boolean logic operators "AND" and "OR" were used to combine search terms. The search string is stated as "Acute kidney injury" OR AKI OR "renal impairment" OR "renal dysfunction" OR "renal disease" AND "Human Immuno-deficiency Virus patients" OR"HIV patients" OR "HIV positive patient*"OR"-Sero-positive patients" OR"Acquired Immune Deficiency Syndrome patients" OR"AIDS patients" OR "people living with HIV/AIDS" OR PLWHA AND Africa (S2 File). Articles were searched by title (ti), abstract (ab), and full text (ft). Modifications of the search results were made by limiters such as study design and country. Articles published up to July 2023 were included in the study. Two reviewers (HMK and CKM) independently searched and screened articles by title, abstract, and full text. The disagreements between the reviewers were resolved through discussion. Further disagreements were solved by the involvement of the third person.

## Inclusion and exclusion criteria

The eligibility criteria of the included studies are summarized in the table below(Table 1).

## Outcome measurement

Acute kidney injury is defined as an increase in serum creatinine of 3 mg/dl within 48 hours and/or 1.5 times the baseline within the previous seven days, as well as urine output of $< 0.5$ mg/kg/hour for six hours [12].

**Table 1. Inclusion and exclusion criteria for included research articles.**

| Criteria | Inclusion criteria | Exclusion criteria |
|---|---|---|
| Population | HIV positive patients | Chronic kidney diseases prior to the exposure of HIV |
| | People living with HIV/AIDS | Patients with dialysis and/or renal dysfunction before the onset of HIV |
| Age | Adult (age $\geq$18 years) | |
| Design | Observational study designs (cross-sectional, cohort, case-control, and survey) | |
| Publication status | Both published and/or unpublished studies such as preprints | Qualitative studies |
| | | Conference papers |
| | | Articles with no full text |
| Country | African countries | Scoping review |
| | | Narrative review |
| | | Systematic review and meta-analysis |
| Publication year | Articles published up-to July 2023 | |
| Language | Any language | |

Notes: AIDS-Acquired Immune-Deficiency Syndrome, HIV-Human Immuno-deficiency Virus

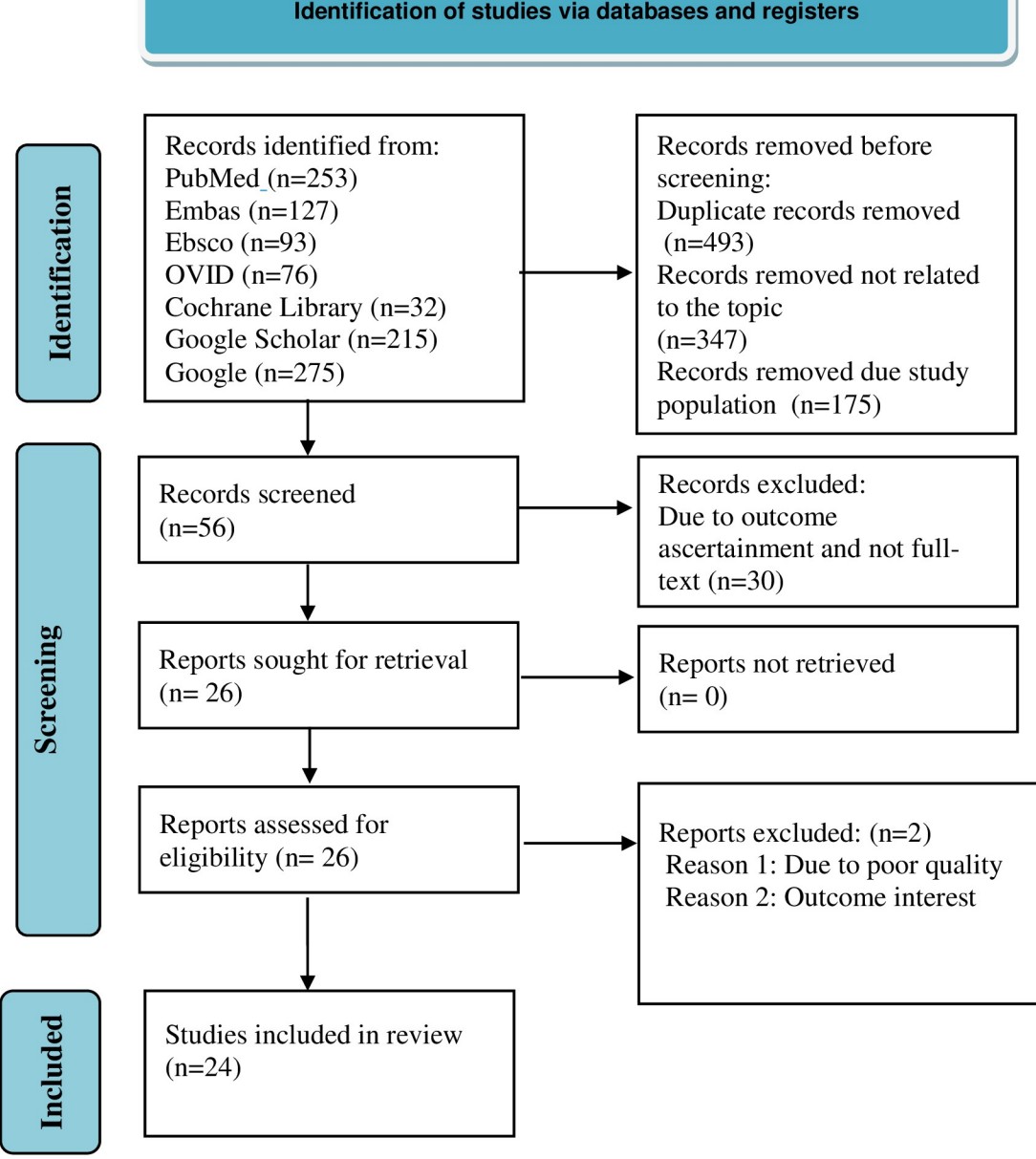

**Fig 1. PRISMA flow chart for flow of information through the phases of systematic review.**

## Data extraction

The data was extracted using a Microsoft Excel spreadsheet. The format was prepared by all the authors and piloted for its clarity, aim, consistency, and depth to capture information from the included articles. Using the data extraction format, information such as author(s), year of publication, study design, country of the study, method of data collection, funding status, sample size, prevalence or incidence of AKI, and effect estimates such as OR with a 95% CI were extracted (S3 File). Two reviewers (HMK and CKM)independently screened the articles by

**Table 2. Characteristics and quality status of included studies.**

| Authors/year | Country | population | Study design | Definition Basis | Data collection | Funding | Sample size | Prevalence (%) | Quality score |
|---|---|---|---|---|---|---|---|---|---|
| Alex MT et al. 2022 [35] | Cameroon | HIV | Cohort | KDIGO -2012 | Lab test, interview | Not reported | 206 | 30.6 | 7 |
| Fiseha T &Gebreweld A. 2021 [19] | Ethiopia | HIV | Cohort | eGFR | Record review | Not funded | 353 | 22.1 | 8 |
| Karoney MJ et al.2022 [34] | Kenya | HIV | Cross-sectional | eGFR | Lab test, interview | EDCTP | 261 | 10 | 9 |
| Kefeni BT, et al. (2021) [20] | Ethiopia | HIV | Cross-sectional | eGFR | Record review, lab test | JU | 352 | 20.7 | 7.5 |
| Kimweri D, et al. (2021) [32] | Uganda | HIV | Cohort | KDIGO | Interview, record review, lab test | not funded | 384 | 19.2 | 8 |
| Mwanja MN et al. (2022) [24] | Tanzania | HIV | Cross-sectional | eGFR | Record review, lab test | TFEL | 396 | 20.7 | 8.5 |
| Mwemezi O, et al. (2020) [25] | Tanzania | HIV | Cross-sectional | EGFR | Record review, lab test | Not funded | 287 | 32.8 | 9 |
| Mugabo C &Ndikubwimana I. (2023) [40] | Rwanda | HIV | Cohort | eGFR | record review | Not reported | 98 | 24.4 | 8.5 |
| Fall K et al. (2017) [38] | Senegal | HIV | Cohort | eGFR | Interview, record review, lab test | not reported | 248 | 12.9 | 8 |
| Nyende L et al. (2020) [33] | Uganda | HIV | Cross-sectional | eGFR | Record review | Mulago KCCA project | 278 | 2.53 | 7 |
| Enyew K, et al. (2016) [21] | Ethiopia | HIV | Cross-sectional | eGFR | lab test | AAU | 60 | 23.3 | 7.5 |
| Mapesi H, et al. (2021) [26] | Tanzania | HIV | Cohort | eGFR | lab test | MOH T | 556 | 7.4 | 7 |
| Vachiat AI, et al. (2013) [36] | South Africa | HIV | cross-sectional | eGFR | Record review | Not reported | 101 | 21 | 8 |
| Ali Y, et al. (2012) [23] | Ethiopia | HIV | Cross-sectional | eGFR | Interview, record review, lab test | Not reported | 321 | 15.9 | 9 |
| Semde A, et al. (2021) [41] | Burkina Faso | HIV | Cross-sectional | KDIGO | Record review | Not reported | 364 | 29.94 | 8.5 |
| Emem CP, et al. (2008) [28] | Nigeria | HIV | Cross-sectional | eGFR | lab test | Not reported | 400 | 38 | 7 |
| Maina SM et al. (2023) [30] | Nigeria | HIV | Case-control | Renal Biopsy | lab test | Not reported | 200 | 10.5 | 7.5 |
| Struik GM, et al. (2015) [37] | Malawi | HIV | Cross-sectional | Cockcroft–Gault equation | lab test | Not reported | 526 | 23.3 | 8 |
| Yilma D, et al. (2012) [22] | Ethiopia | HIV | Cross-sectional | eGFR | lab test | FAD | 340 | 26.9 | 9 |
| Okafor UH, et al. (2015) [29] | Nigeria | HIV | Cross-sectional | eGFR | Interview, record review, lab test | Not reported | 383 | 53.3 | 9 |
| Sakajiki MA, et al. (2011) [31] | Nigeria | HIV | Cross-sectional | eGFR | Interview, record review, lab test | Not reported | 240 | 56.8 | 7.5 |
| Kilonzo BS, et al. (2016) [27] | Tanzania | HIV | Cross-sectional | eGFR | Interview, record review, lab test | Not reported | 637 | 28 | 8 |
| Tembo S, et al. (2017) [39] | Zambia | HIV | Cross-sectional | eGFR | lab test | Not reported | 360 | 24.76 | 9 |
| Ekat MH, et al. (2012) [42] | DR Congo | HIV | Cross-sectional | eGFR | lab test | No reported | 562 | 8.5 | 8.5 |

**Notes**:- EDC: European and Developing Countries, eGFR: estimated Glomerular Filtration Rate, KDIGO: Kidney disease Improving Global Outcome, HIV: Human immunodeficiency virus; TFEL-Tanzania Field Epidemiology and Laboratory, MOH T- Ministry of Health Tanzania, FAD-Foreign affairs of Denmark

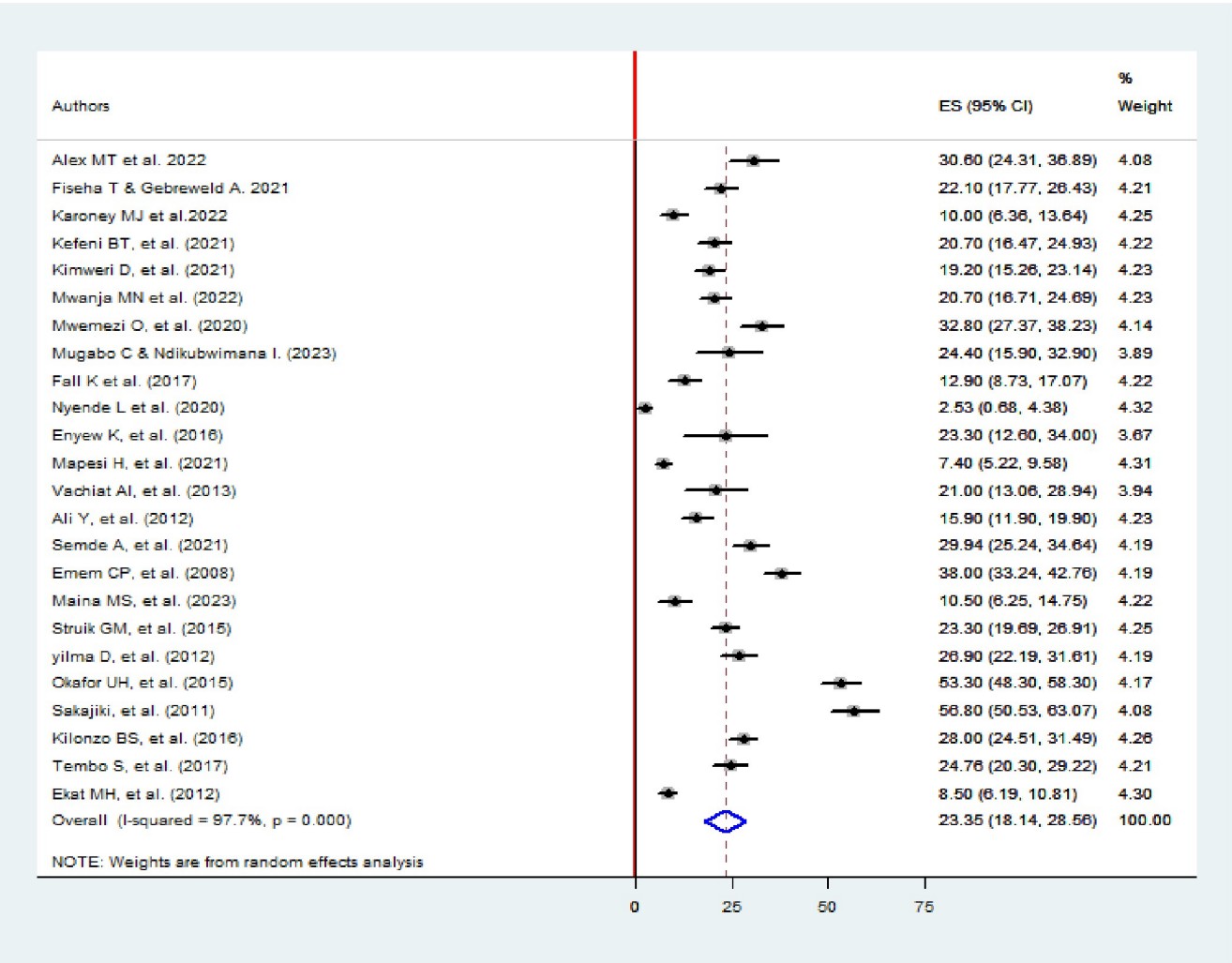

**Fig 2. The forest plot shows the pooled prevalence of AKI among HIV patients in Africa.**

titles, abstracts, and full texts. Possible inconveniencies were solved by discussion with the reviewers and/or the involvement of the third person.

## Quality appraisal

Articles were assessed for their quality using the Newcastle-Ottawa assessment scale adapted from cross-sectional [13], cohort [14], and case-control studies [15]. A score of 6 or above was considered a high-quality article. Two reviewers (AWA and YMF) assessed the quality of the articles. The reviewers compared the quality of the appraisal scores and resolved inconsistencies before calculating the final appraisal score.

## Data analysis

The extracted data was exported to Stata version 14 for analysis. Heterogeneity was detected by Cochrane Q statistics and $I^2$. The heterogeneity test statistics results of below 25%, 50%, and

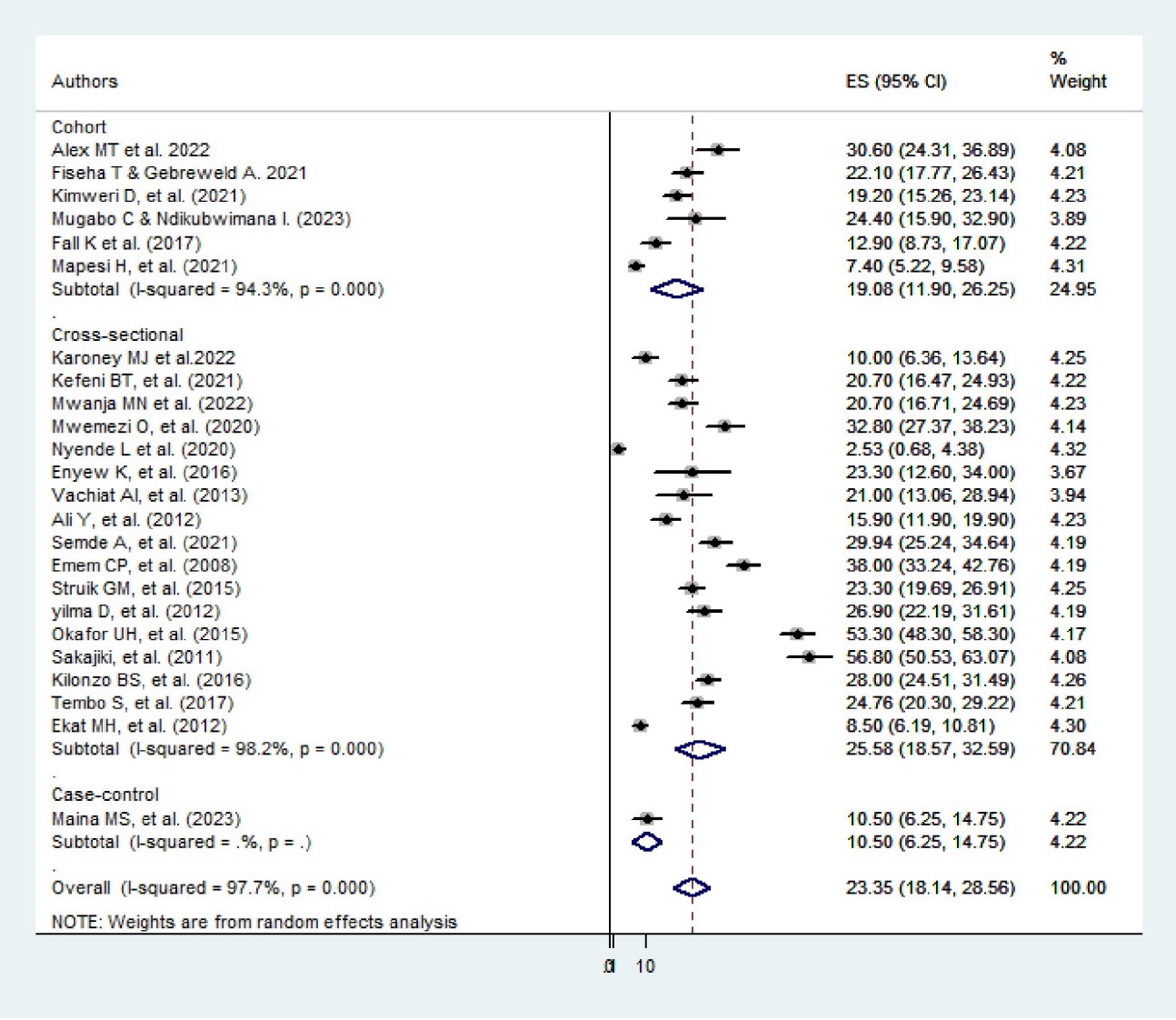

**Fig 3. Subgroup analysis by study design among the included studies.**

above 75% were declared as low, moderate, and high heterogeneity [16], respectively. A random effect meta-analysis model was used to estimate the pooled prevalence of AKI [17]. The graphic asymmetry of the funnel plot test and/or Egger's test (p-value <0.005) were used to detect the publication bias [18].

**Table 3. Meta-regression analysis for variation of the included studies.**

| Variables | Std.err | p-value | Coef. | 95%CI |
|---|---|---|---|---|
| Publication year | 0.584 | 0.174 | 0.82 | 0.39–1.203 |
| Sample size | 0.345 | 0.001 | 3.11 | 2.399–3.83 |

**Notes:** CI-confidence interval, Coef.-coefficient, Std.err- Standard error

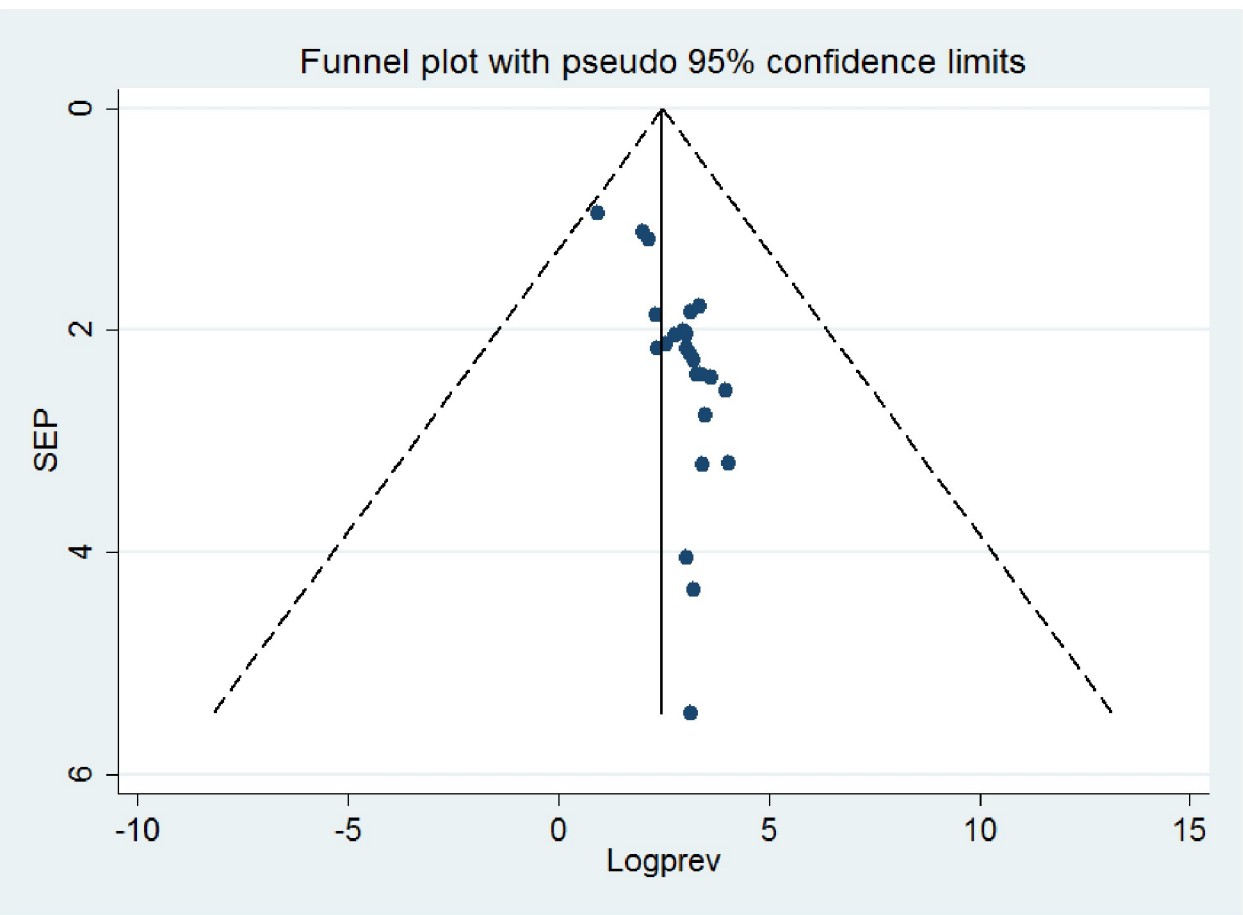

**Fig 4. Funnel plot to assess the publication bias of the included studies.**

## Results

### Study selection and characteristics of included studies

The search strategy retrieved 1071 original articles. About 493, 347, and 175 articles were removed due to duplication, not related to the topic of interest, and population variation, respectively. About 56 articles remained, and following further screening, 30 articles were removed due to variations in outcome ascertainment and not full-text. Then 26 full-text articles were assessed for eligibility, of which two articles were excluded because of poor quality and outcome interest (Fig 1). Finally, twenty-four articles were retrieved and included in the review, with a total of 9713 populations. Of the twenty-four studies, five were from Ethiopia [19–23], four from Tanzania [24–27], four from Nigeria [28–31], two from Uganda [32, 33], one from Kenya [34], one from Cameroon [35], one from South Africa [36], one from Malawi [37], one from Senegal [38], one from Zambia [39], one from Rwanda [40], one from Burkina Faso [41], and one from DR Congo [42]. The pooled prevalence was calculated from the aforementioned studies, whereas for predictors of AKI, three studies for hemoglobin level [19, 35, 41], six studies for CD4 count [19, 20, 24–26, 31], and three studies for WHO clinical HIV staging [20, 26, 35] were used. The prevalence of AKI among HIV patients ranged from 2.53% in Uganda [33] to 56.8% in Nigeria [31]. The majority of the studies used lab tests as a method of data collection. All of the included studies had high-quality scores(Table 2).

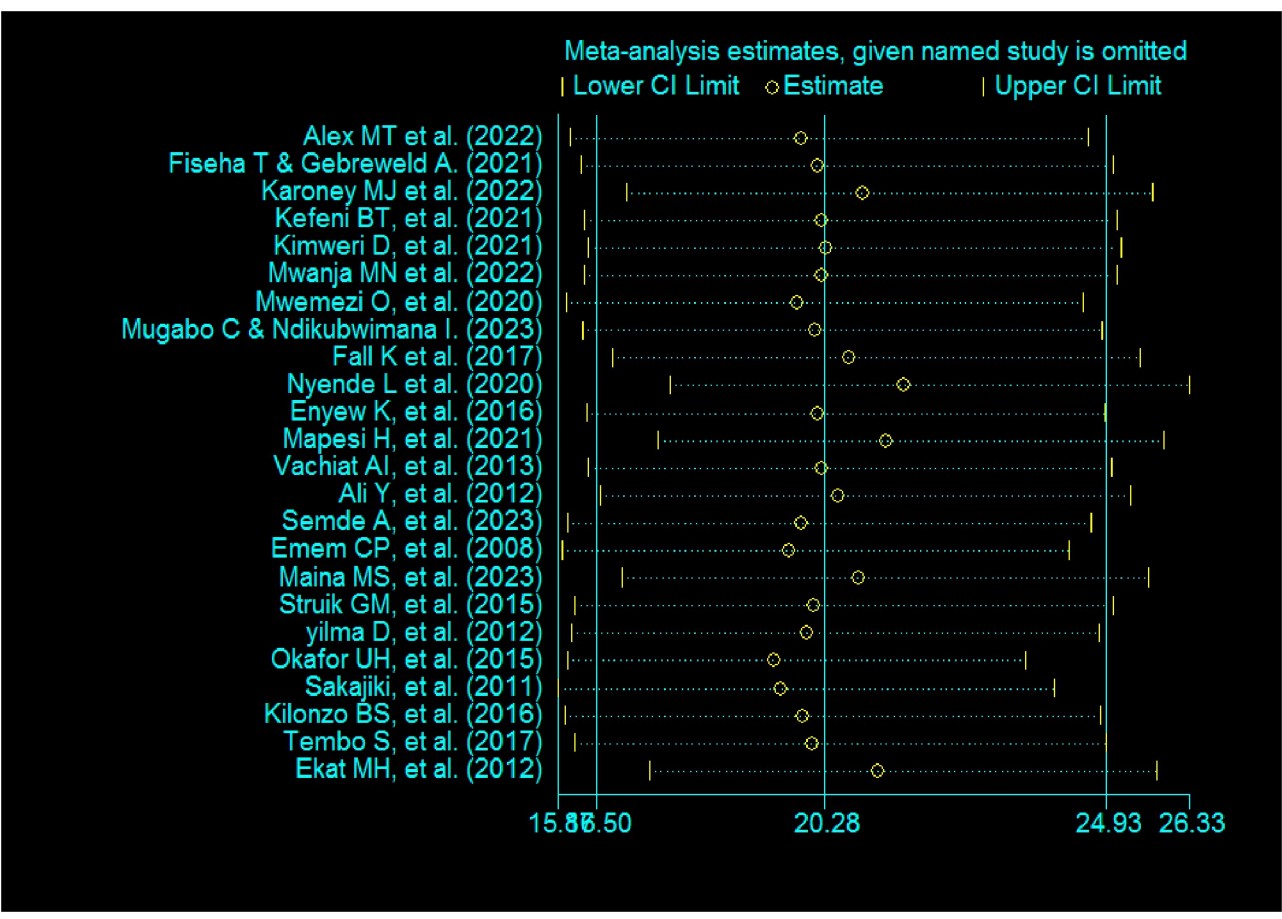

**Fig 5. Sensitivity analysis among the included study to detect the effect of single study.**

## The prevalence of acute kidney injury

The pooled prevalence of AKI among HIV patients was found to be 23.35% (95% CI: 18.14–28.56%, $I^2$ = 97.7%, p-value <0.001) (Fig 2).

## Subgroup analysis

The subgroup analysis was computed using the study designs of the included studies. In the subgroup analysis, the level of heterogeneity is high. In the cohort designs, the level of heterogeneity was 94.3%, whereas in cross-sectional designs it was 98.2% (Fig 3). Therefore, meta-regression analyses need to be computed to identify the real source of the variations.

## Meta-regression analysis

To identify the causes of the covariates, both bivariate and multivariate meta-regression analyses were computed. In the multivariate analyses, sample size was found to be the cause of the variations (AOR = 3.11, 95%CI: 2.399–3.83) (Table 3).

## Publication bias

The publication bias was assessed by the graphic asymmetry test of the funnel plot and/or Egger's tests. The funnel plot test showed that there was an asymmetrical distribution,

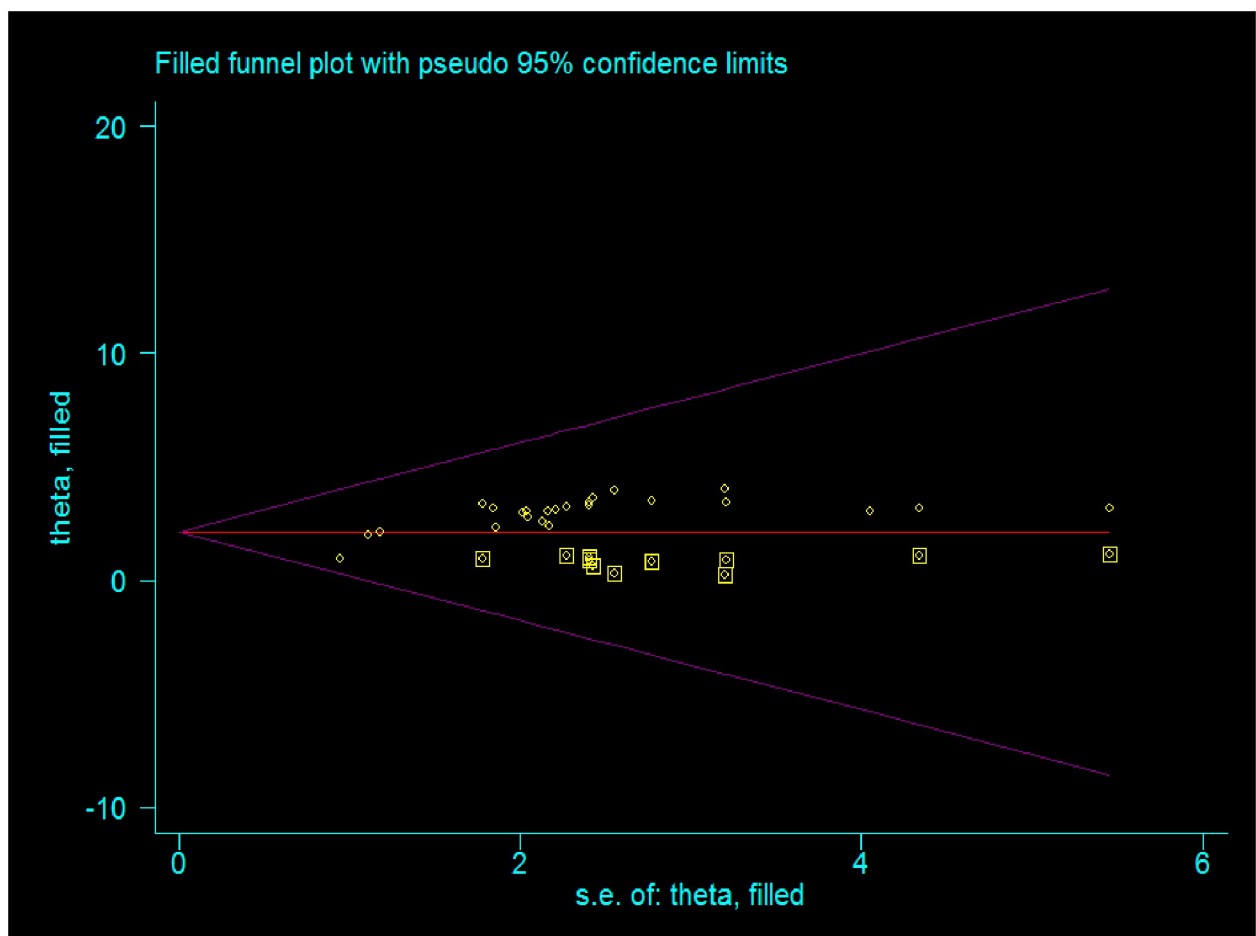

**Fig 6. Trim and fill analysis results among articles.**

indicating there is a publication bias (Fig 4). Similarly, Egger's test shows that the p-value is <0.001, indicating there is publication bias. Sensitivity analysis was computed to see the effect of a single study on the summary effect estimates, indicating there is no single study effect or outlier (Fig 5). To treat small study effects, non-parametric trim and fill analyses were computed. As a result, eleven articles were filled, making a total of 35 articles with 34 degrees of freedom, a p-value of 1.00, and a moment-based estimate of between-study variance of 0.00 (Fig 6).

## Predictors of AKI among HIV patients

**Hemoglobin level.** In this review study, having low hemoglobin (hgb<8mg/dl) was found to be the determinant factor of AKI among HIV-positive patients (AOR = 2.4; 95% CI:1.69–3.4, $I^2$ = 0.0%, p-value = 0.40).From the forest plot, the $I^2$ was 0.0% and the p-value was 0.40 (Fig 7), indicating there is no variation across the studies.

**CD4 count.** The findings of this study revealed that a low CD4 count is not a risk factor for AKI among HIV patients(AOR = 2.25; 95% CI: 0.85–5.93, $I^2$ = 0.0%, p-value = 0.847). As shown in the figure below, the $I^2$ was 0.0% and the p-valuewas0.847, meaning that there was no variation across the included studies (Fig 8).

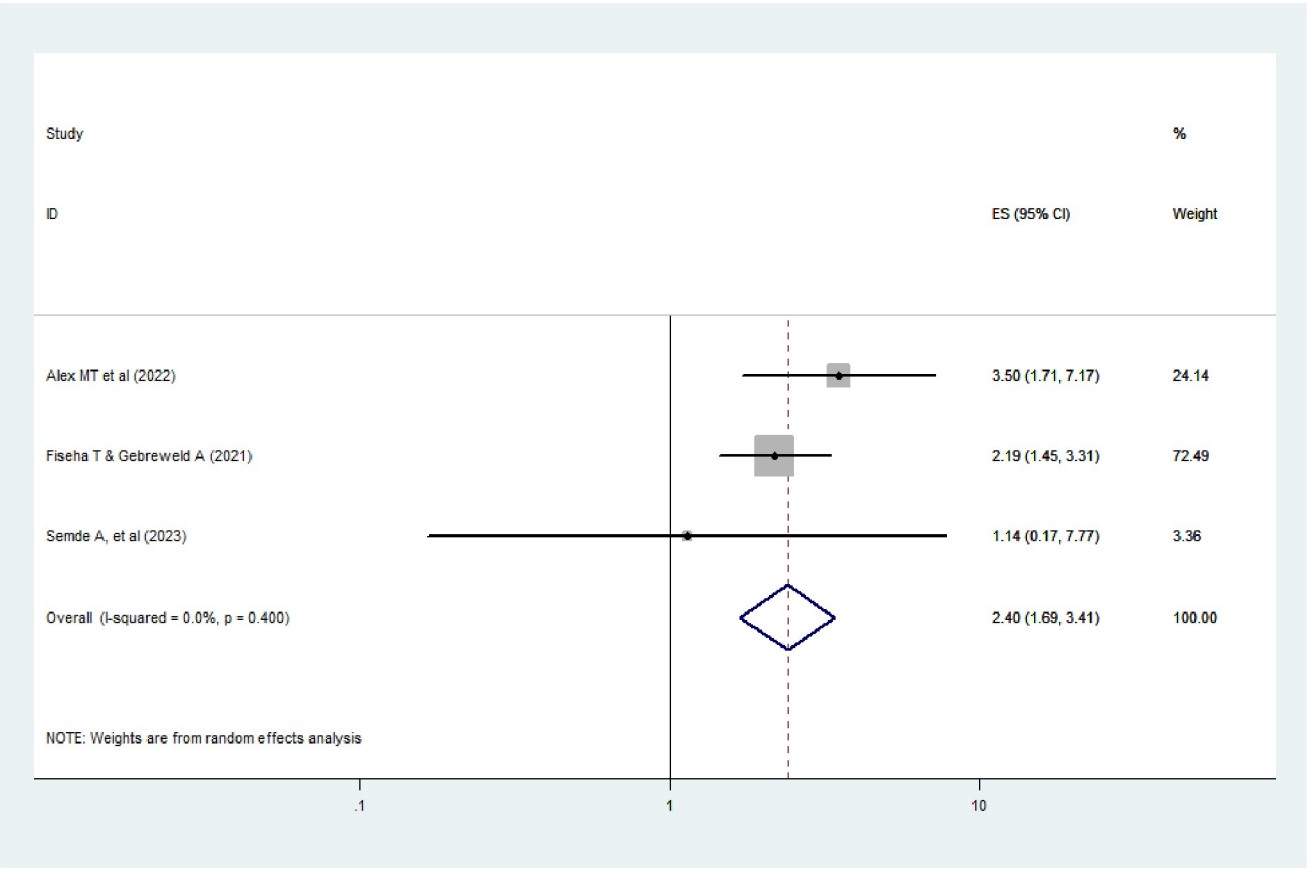

**Fig 7. The effect of hemoglobin on AKI among HIV positive patients.**

**WHO clinical HIV staging.** The study depicts that the WHO clinical HIV stage is not a predictor for AKI among HIV-positive patients(AOR = 2.29, 95% CI: 0.49–10.74, $I^2$ = 0.0%, p-value = 0.955) (Fig 9). From the forest plot, the $I^2$ was 0.00% and the p-value was 0.955, which indicates there is no variation across the study.

## Discussion

The findings of this review study revealed that the pooled prevalence of AKI among HIV-positive patients in Africa was 23.35% (95% CI: 18.14–28.56%). The result of this study is lower than the studies conducted in Shanghai, China 12.5% [43] and 0.7%Edmonton, Canada [44]. The possible reason for the discrepancy is that the Chinese study focused on incidence and used a variety of outcome measurement criteria, whereas the Canadian study focused on the effect of Tenofovir Disoproxil Fumarate (TDF) on the development of AKI and used clinical trial studies, whereas the current study used observational studies.

A low hemoglobin level was found to be the determinant factor of AKI among HIV-positive patients. HIV-positive patients who had a low hemoglobin level (hgb<8mg/dl) were2.4 times more likely to develop AKI compared to normal hemoglobin levels. This is supported by a study in Seoul [45], which stated that the odds of AKI are associated with decreased hemoglobin. The possible reasons that renal insufficiency is associated with anemia (low hemoglobin)

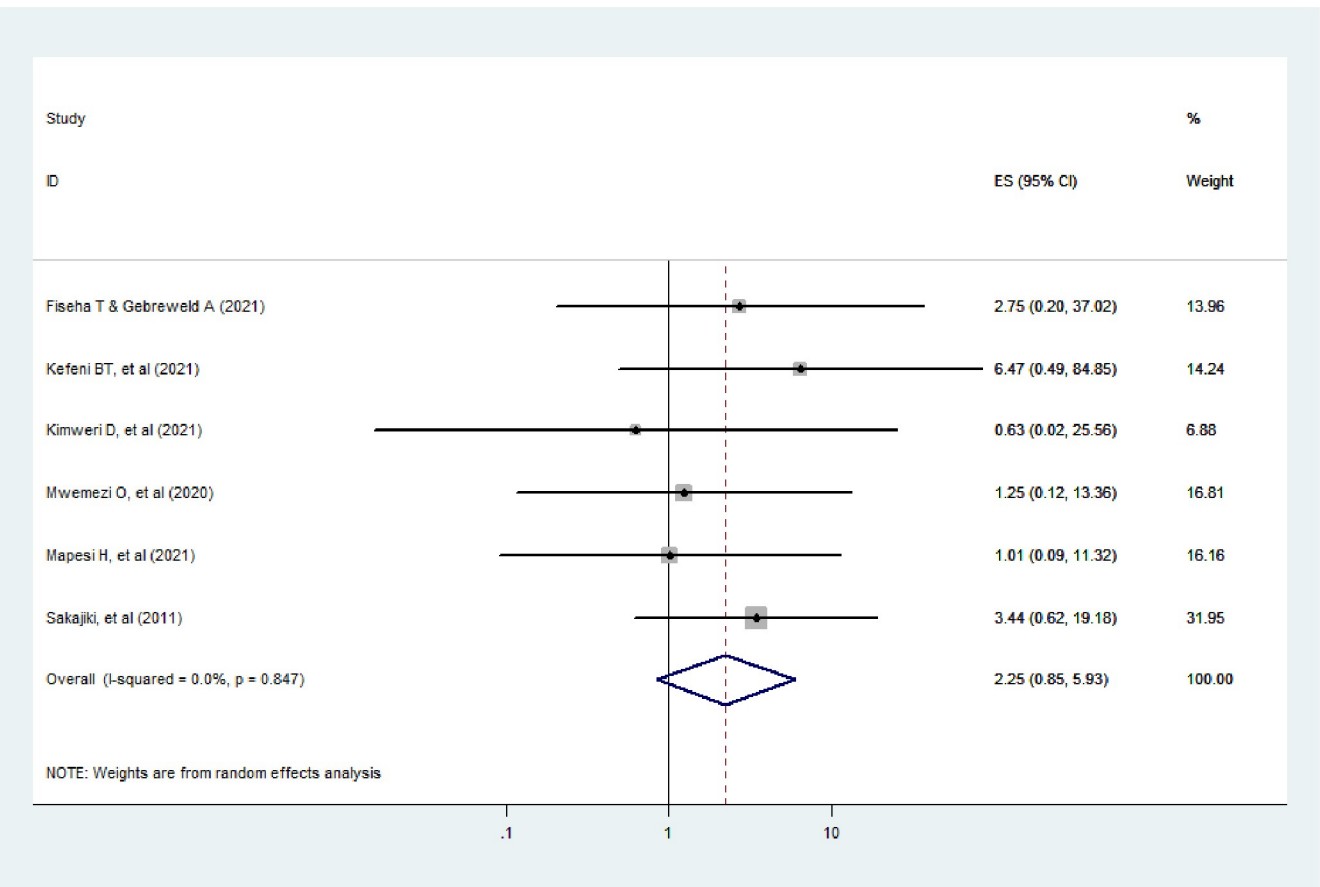

**Fig 8. The effect of CD4 count on AKI among HIV positive patients.**

[46]. A low hemoglobin level decreases renal tissue oxygenation and results in acute tubular necrosis.

The sample size was one of the possible sources of variation across the studies. In the meta-regression analysis, sample size is the source of the variation for the included studies. This is due to the fact that when the sample size is too small, the bias from sampling will increase the frequency of sampling [47]. Larger sample sizes are required to attain the desired normal power [48]. Additionally, the small sample size resulted ina small effect size, which can lead to the possibility of variation.

As a strength, the authors investigate HIV-associated AKI in Africa at large. In addition, the study investigates both the prevalence and associated factors of AKI among HIV-positive patients.

The study has the following limitations: Firstly, the study did not assess the stage of AKI, mortality, or its treatment outcome. Therefore, we recommendedthat further global researches which need to be carried out to estimate the global prevalence and identify other important predictors of HIV-associated AKI among HIV-positive populations.

## Implications of the study

The findings of this study provide input for healthcare workers to provide an integrated HIV care service along with routine ART services. The study gives direction for healthcare workers

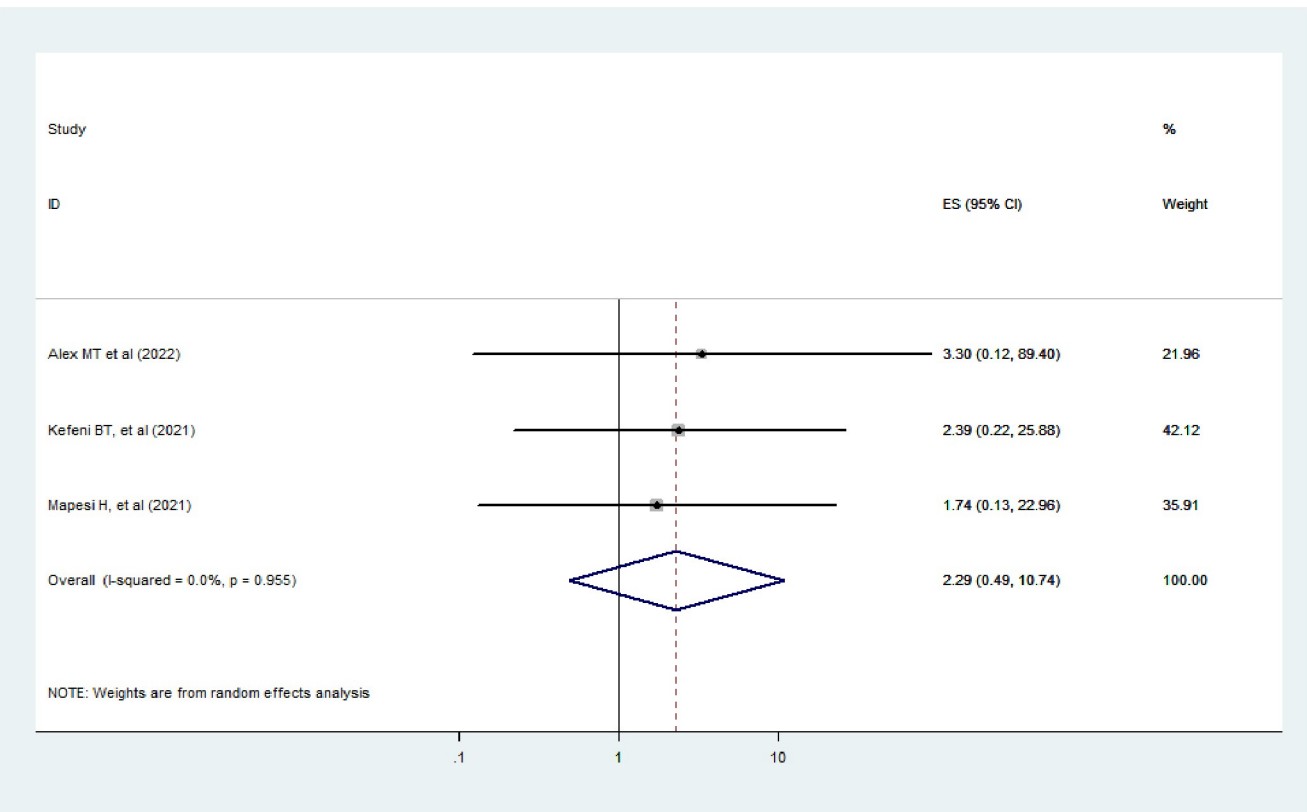

**Fig 9. The effect of WHO clinical staging on AKI among HIV positive patients.**

to emphasize on regular monitoring of renal function tests to HIV-positive patients. Additionally, the study also helps policy makers and decision-makers focus on intervention strategies for the early detection and prevention of AKI in HIV positives.

## Conclusions

This systematic review and meta-analysis revealed that the pooled prevalence of AKI among HIV-positive patients in Africa was high. HIV-positive patients with low hemoglobin levels are at risk for developing AKI. Hence, regular monitoring of kidney function tests is needed to prevent, detect, and treat AKI. Therefore, healthcare workers should provide integrated HIV-renal healthcare services such as renal function tests to prevent, detect, and treat AKIto reduce its progression to advanced stages and complications.

## Supporting information

**S1 File. PRISMA checklist.**
(DOCX)

**S2 File. Search strategy.**
(DOCX)

**S3 File. Data availability statement.**
(DOCX)

## Acknowledgments

We would like to acknowledge the team members for their invaluable contribution from the conception to the final approval for submission to publication.

## Author Contributions

**Conceptualization:** Abere Woretaw Azagew.

**Data curation:** Abere Woretaw Azagew, Hailemichael Kindie Abate, Yohannes Mulu Ferede, Chilot Kassa Mekonnen.

**Formal analysis:** Abere Woretaw Azagew, Hailemichael Kindie Abate, Yohannes Mulu Ferede, Chilot Kassa Mekonnen.

**Investigation:** Abere Woretaw Azagew, Hailemichael Kindie Abate, Yohannes Mulu Ferede, Chilot Kassa Mekonnen.

**Methodology:** Abere Woretaw Azagew, Hailemichael Kindie Abate, Yohannes Mulu Ferede, Chilot Kassa Mekonnen.

**Project administration:** Abere Woretaw Azagew, Hailemichael Kindie Abate, Yohannes Mulu Ferede, Chilot Kassa Mekonnen.

**Resources:** Abere Woretaw Azagew, Hailemichael Kindie Abate, Yohannes Mulu Ferede, Chilot Kassa Mekonnen.

**Software:** Abere Woretaw Azagew, Hailemichael Kindie Abate, Yohannes Mulu Ferede, Chilot Kassa Mekonnen.

**Supervision:** Abere Woretaw Azagew, Hailemichael Kindie Abate, Yohannes Mulu Ferede, Chilot Kassa Mekonnen.

**Validation:** Abere Woretaw Azagew, Hailemichael Kindie Abate, Yohannes Mulu Ferede, Chilot Kassa Mekonnen.

**Visualization:** Abere Woretaw Azagew, Hailemichael Kindie Abate, Yohannes Mulu Ferede, Chilot Kassa Mekonnen.

**Writing – original draft:** Abere Woretaw Azagew, Hailemichael Kindie Abate, Yohannes Mulu Ferede, Chilot Kassa Mekonnen.

**Writing – review & editing:** Abere Woretaw Azagew, Hailemichael Kindie Abate, Yohannes Mulu Ferede, Chilot Kassa Mekonnen.

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
