## [Decision Letter · Decision Letter 0]

29 Aug 2023

PONE-D-23-23314Acute Kidney Injury and Its predictors among HIV-positive Patients in Africa: Systematic review and Meta-analysisPLOS ONE

Dear Dr. Azagew,

Thank you for submitting your manuscript to PLOS ONE. After careful consideration, we feel that it has merit but does not fully meet PLOS ONE’s publication criteria as it currently stands. Therefore, we invite you to submit a revised version of the manuscript that addresses the points raised during the review process.

We look forward to receiving your revised manuscript.

Kind regards,

Palesa Motshabi Chakane, PhD

Academic Editor

PLOS ONE

Journal Requirements:

Additional Editor Comments :

The authors are requested to review their data search strategy separate from the inclusion and exclusion criteria. It is not clear the characteristics of manuscripts they included in their data search. The authors are also requested to share an extensive search strategy that they used (for any one of the databases they used). Further review will only be performed with these requirements.

Reviewers' comments:

Reviewer's Responses to Questions

**Comments to the Author**

1. Is the manuscript technically sound, and do the data support the conclusions?

Reviewer #1: Partly

2. Has the statistical analysis been performed appropriately and rigorously? 

Reviewer #1: I Don't Know

3. Have the authors made all data underlying the findings in their manuscript fully available?

Reviewer #1: No

4. Is the manuscript presented in an intelligible fashion and written in standard English?

Reviewer #1: Yes

5. Review Comments to the Author

Reviewer #1: Thank you for the paper. I just wanted to find some clarity with regards to the following:

- the PECO/PEO framework?

- lines 116-119: AKI is defined according to AKIN criteria; however, it is not clear from the extracted/ or included articles if they all utilized AKIN in their original articles

- In terms of other confounding factors in the patients included what type of ARTs were they on? Any other medications and/ or comorbidities? who ended up on RRT?

- Other variables: why was Hemoglobin, CD4 count and WHO clinical staging only variables chosen?

- Consistency with the referencing (others in CAPITAL LETTERS)

Good Luck

6. PLOS authors have the option to publish the peer review history of their article (what does this mean?). If published, this will include your full peer review and any attached files.

Reviewer #1: **Yes: **Gontse Leballo

---

## [Author Response · Author response to Decision Letter 0]

5 Oct 2023

The response to the reviewers attached as file attachment section. Due to internet blockage by the Government of Ethiopia, I can't access it as usual, so I kindly provide the privilege to the editors to correct errors associated with editing problems.

---

## [Decision Letter · Decision Letter 1]

21 Nov 2023

PONE-D-23-23314R1Acute Kidney Injury and Its predictors among HIV-positive patients in Africa: Systematic review and meta-analysisPLOS ONE

Dear Dr. Azagew,

Thank you for submitting your manuscript to PLOS ONE. After careful consideration, we feel that it has merit but does not fully meet PLOS ONE’s publication criteria as it currently stands. Therefore, we invite you to submit a revised version of the manuscript that addresses the points raised during the review process.

We look forward to receiving your revised manuscript.

Kind regards,

Palesa Motshabi Chakane, PhD

Academic Editor

PLOS ONE

Journal Requirements:

Additional Editor Comments:

The paper is a great addition to the literature on acute kidney injury. The authors are applauded on highlighting this disease process in HIV positive patients in Africa. The systematic review was performed in an applaudable manner. The authors are requested to submit the paper to a language editor. The paper is accepted with minor revision as indicated.

Reviewers' comments:

Reviewer's Responses to Questions

**Comments to the Author**

1. If the authors have adequately addressed your comments raised in a previous round of review and you feel that this manuscript is now acceptable for publication, you may indicate that here to bypass the “Comments to the Author” section, enter your conflict of interest statement in the “Confidential to Editor” section, and submit your "Accept" recommendation.

Reviewer #1: All comments have been addressed

2. Is the manuscript technically sound, and do the data support the conclusions?

Reviewer #1: Yes

3. Has the statistical analysis been performed appropriately and rigorously? 

Reviewer #1: Yes

4. Have the authors made all data underlying the findings in their manuscript fully available?

Reviewer #1: Yes

5. Is the manuscript presented in an intelligible fashion and written in standard English?

Reviewer #1: Yes

6. Review Comments to the Author

Reviewer #1: Thank you for a well presented research paper. Hoping that the results of this systematic review will encourage prospective work from the African perspective.

7. PLOS authors have the option to publish the peer review history of their article (what does this mean?). If published, this will include your full peer review and any attached files.

Reviewer #1: **Yes: **Gontse Leballo

---

## [Author Response · Author response to Decision Letter 1]

28 Nov 2023

The whole manuscript edited and corrected by professional language editor.

---

## [Decision Letter · Decision Letter 2]

23 Jan 2024

Acute kidney injury and its predictors among HIV-positive patients in Africa: Systematic review and Meta-analysis

PONE-D-23-23314R2

Dear Dr. Abere Woretaw Azagew

We’re pleased to inform you that your manuscript has been judged scientifically suitable for publication and will be formally accepted for publication once it meets all outstanding technical requirements.

Kind regards,

Palesa Motshabi Chakane, PhD

Academic Editor

PLOS ONE

Additional Editor Comments (optional):

Reviewers' comments:

Reviewer's Responses to Questions

**Comments to the Author**

1. If the authors have adequately addressed your comments raised in a previous round of review and you feel that this manuscript is now acceptable for publication, you may indicate that here to bypass the “Comments to the Author” section, enter your conflict of interest statement in the “Confidential to Editor” section, and submit your "Accept" recommendation.

Reviewer #1: All comments have been addressed

2. Is the manuscript technically sound, and do the data support the conclusions?

Reviewer #1: Yes

3. Has the statistical analysis been performed appropriately and rigorously? 

Reviewer #1: Yes

4. Have the authors made all data underlying the findings in their manuscript fully available?

Reviewer #1: Yes

5. Is the manuscript presented in an intelligible fashion and written in standard English?

Reviewer #1: Yes

6. Review Comments to the Author

Reviewer #1: - The Authors have addressed all the concerns by the reviewers.

- The manuscript is both novel and scientifically presented. The results make sense and are well presented.

Satisfied with the scientific paper. Thank you.

7. PLOS authors have the option to publish the peer review history of their article (what does this mean?). If published, this will include your full peer review and any attached files.

Reviewer #1: **Yes: **Gontse Leballo

---

## [Editor Report · Acceptance letter]

1 Feb 2024

PONE-D-23-23314R2 

PLOS ONE

Dear Dr. Azagew, 

I'm pleased to inform you that your manuscript has been deemed suitable for publication in PLOS ONE. Congratulations! Your manuscript is now being handed over to our production team.

Kind regards, 

on behalf of

Dr. Palesa Motshabi Chakane 

Academic Editor

PLOS ONE